# Effectiveness of radiofrequency and exercise-based rehabilitation on symptoms associated with pelvic floor dysfunction in breast cancer patients: A study protocol

Cristina Salar-Andreu[1], Sergio Montero-Navarro[1], Ana Lozano-Rubio[1], Sonia Del Rio-Medina[1], Jose M. Botella-Rico[1], María Torres-Lacomba[2,3,4], Josep C. Benítez-Martínez [5], Jesús Sánchez-Más [6]*, Cristina Orts-Ruiz[1]

1 Physical Therapy Department, Health Sciences Faculty, CEU-Cardenal Herrera University, CEU Universities, Elche, Valencia, Spain, 2 Faculty of Medicine and Health Sciences, Physiotherapy and Nursing Department, University of Alcalá, Madrid, Spain, 3 Physiotherapy in Women's Health Research Group, University of Alcalá, Madrid, Spain, 4 Ramón y Cajal Institute of Health Research - IRYCIS, Hospital Universitario Ramón y Cajal, Madrid, Spain, 5 Research Group on Physiotherapy Technology and Recovering (FTR), University of Valencia, Valencia, Spain, 6 Biomedical Sciences Department, Health Sciences Faculty, CEU-Cardenal Herrera University, CEU Universities, Elche, Valencia, Spain

☙ These authors contributed equally to this work.
* jesus.sanchez@uchceu.es

## Abstract

### Background

Breast cancer is currently the most diagnosed type of cancer in the world, with a five-year survival rate of 90%. Survivors develop genitourinary dysfunction symptoms due to cancer treatment, which implies that they have to endure physical and psychological sequelae, with a negative impact on their quality of life. We present a study protocol to verify the effect of radiofrequency (RF) and pelvic floor muscle training (PFMT) for the treatment of genitourinary dysfunction in breast cancer survivors.

### Methods/Design

This is a double-blind, three-arm, randomised clinical trial (Registration: NCT06694519). Participants from two breast cancer associations from Alicante (Spain) will be randomly assigned to one of the three intervention groups (RF, PFMT, RF + PFMT). It will include survivors aged ≥ 18 years who present pelvic dysfunction assessed by the Pelvic Floor Distress Inventory questionnaire (PFDI20) ≥ 100. Pelvic muscle strength, pelvic function, vaginal symptoms, sexual function, self-esteem and quality of life will be evaluated before the intervention, with follow-up at 15 days, 6 months and 1 year after the intervention.

**Data availability statement:** No datasets were generated or analysed during the current study. All relevant data from this study will be made available upon study completion.

**Funding:** The author(s) received no specific funding for this work.

**Competing interests:** The authors declare that there are competing financial interests in relation to the described work associated with the funding received by the company Capenergy Medical (Barcelona, Spain). Capenergy Medical (Barcelona, Spain) will assign the device for applying RF with capacitive electrical transfer (Capernergy® device model C500 Urogyne). This does not alter our adherence to PLOS ONE policies on sharing data and materials. There are no other conflicts of interest.

## Expected results

RF could offer additional benefits to PFMT due to its proven effectiveness in the treatment of vaginal dryness and dyspareunia. The expected results will have a positive impact on the health and well-being of women with breast cancer, reducing the symptomatology associated with the disease and its treatment, and improving their quality of life, as well as providing value for the development of more effective treatment protocols.

## Trial registration

ClinicalTrials.gov NCT06694519

## Introduction

Breast cancer is currently the most diagnosed cancer in the world (about 2.3 million in 2022) and is the leading cause of cancer mortality in women (666,103 in 2022) [1]. Spain registered 40,203 new cases of breast cancer in 2023. It has a five-year survival rate of 90% [2, 3], which increases by about 0.5% every year, as a result of the implementation of early cancer detection programmes and the evolution of comprehensive treatments [4]. However, due to the disease itself and the treatments, breast cancer survivors suffer physical, functional and psychological sequelae that have a negative impact on their physical and emotional well-being, negatively affecting their quality of life [5, 6].

The treatment of breast cancer varies depending on the patient's individual characteristics, the subtype of the disease and its extent, whether limited to the breast and lymph nodes or has spread to other parts of the body. A combination of medical treatments are used to reduce the likelihood of cancer recurrence, such as surgery for breast tumour removal (partial or total mastectomy, complete axillary dissection or sentinel node biopsy), radiation therapy to reduce the risk of recurrence in the breast tissues and surrounding regions and pharmacological treatments aimed at eliminating cancer cells and preventing their spread, which may include hormonal therapies, chemotherapy or specific treatments with biological products (immunotherapy) [7,8]. However, these treatments are associated with side effects including dysfunctions in women's sexual life, infertility, anovulation, amenorrhoea, vaginal atrophy, hot flushes and early menopause [9–12].

The genitourinary syndrome of menopause (GSM) is common in 50% of menopausal women, but its incidence is higher in women with breast cancer, mainly due to surgery, chemotherapy, radiation and hormone therapy [10]. GSM is characterised by vaginal dryness, vulvar and vaginal burning and irritation, lack of lubrication, dyspareunia, dysuria, urinary urgency and recurrent urinary tract infections [10,13], causing a decrease in sexual interest and desire [14]. GSM is also associated with pelvic floor and intestinal tract dysfunctions such as constipation and diarrhoea [10,15]. In this regard, it has been described that between 40–59% of women with breast cancer have symptoms of pelvic floor dysfunction [13,16], with urinary incontinence (UI)

being the most common genitourinary dysfunction among women with breast cancer, followed by faecal incontinence and, to a lesser extent, prolapse [17–19].

Various treatment options are available for GSM-associated pelvic dysfunction. Invasive options include abdominal or vaginal surgery [20–22] and urethral bulking by injecting synthetic materials into the urethral mucosa to provide support and narrow the opening of the bladder neck [23]. The most common post-surgical complications include difficulty in urination, urinary tract infection, postoperative dyspareunia, mesh erosion, persistence or recurrence of UI or pelvic organ prolapse, genitourinary tract injury (including bladder perforation) and gastrointestinal tract injury [24,25]. Among the approved pharmacological options, oxybutynin is notable, an anticholinergic drug that blocks the muscarinic receptor in the smooth muscle of the bladder and inhibits detrusor muscle contraction. It usually causes dry mouth and constipation as predominant side effects [26].

The conservative approach includes various interventions aimed at optimising the patient's lifestyle, the normalisation of the pelvic floor muscles, as well as other strategies such as bladder re-education, electrostimulation, radiofrequency or the use of moisturising gels and lubricants. Changes in lifestyle and regular physical exercise are the first-line treatments to mitigate menopausal symptoms [27]. The therapeutic physical exercise approach includes CORE normalisation (stabilising muscles of the abdominal and lumbopelvic girdle) and pelvic floor muscle training (PFMT). Programmes should encompass muscle strengthening exercises, stretching, postural techniques, abdominal diastasis treatment and pelvic floor re-education. Strengthening the pelvic floor muscles and improving proprioception are the benefits attributed to physical exercise responsible for improving abdominal-pelvic symptoms, sexual function and quality of life [28–31]. PFMT in post-menopausal women improves the symptoms and signs of GSM-associated pelvic dysfunction, with positive impact on activities of daily living, quality of life and sexual function [32–34].

RF is another conservative technique that has gained recognition due to its practicality and feasibility as a safe and effective therapy to treat vaginal symptoms and orgasmic dysfunction and stress UI (SUI) [35,36]. This technique, which uses high-frequency electromagnetic waves to generate thermal microdots in the superficial and deep dermis, which allows tissue regeneration by promoting the production and reorganisation of collagen, elastic fibres and vascularisation, has been effective as a treatment for vaginal dryness and dyspareunia, improving trophism in post-menopausal patients with GSM symptoms [37, 38]. A recent study conducted in climacteric women with SUI showed that conservative therapy combined RF and PFMT led to a significant improvement in SUI symptoms and vaginal dryness [39]. To date, no clinical trial has evaluated the role of RF as a treatment for symptoms associated with the typical GSM-related pelvic dysfunction in patients with breast cancer. Due to the proven safety and efficacy of RF as a conservative therapy for pelvic dysfunction, the general objective will be to compare the effect of RF and PFMT on GSM-associated symptoms in patients with breast cancer. The specific objectives are related to the evaluation of pelvic floor muscle dysfunction (UI, prolapse and colorectal function). The Vaginal Health Index and the Sexual Distress Scale will also be evaluated. We will ultimately evaluate the impact of the treatment on quality of life and self-perceived body image.

Our hypothesis is that conservative RF-based treatment will act as an adjuvant therapy for PFMT that will improve the symptoms associated with the onset of GSM and pelvic function in women diagnosed with breast cancer, and therefore improve the quality of life of cancer survivors.

## Materials and methods

### Trial design

This is a randomised, three-arm, double-blind clinical trial. The follow-up involves four evaluations: one prior to the intervention, and three after the end of the intervention: at fifteen days, at six months and at twelve months. Participants will be randomly assigned to one of the three intervention groups (RF, PFMT or RF+PFMT). The study will follow the guidelines set according to CONSORT [40] and the SPIRIT Statement [41] (S1 File). Fig 1 describes the study in detail.

| | STUDY PERIOD | | | | | | | | |
|---|---|---|---|---|---|---|---|---|---|
| | Enrolment | Baseline | Post-allocation: Interventions, Follow-up | | | | | | |
| TIMEPOINT | 5 May 2025 to 31 July 2025 | 0 w | 1-4 w | 5-8 w | 9-12 w | 13-16 w | F1 | F2 | F3 |
| **ENROLMENT:** | | | | | | | | | |
| Eligibility criterio | X | | | | | | | | |
| Recruitment Initial | X | | | | | | | | |
| Assessment | X | | | | | | | | |
| Informed consent | X | | | | | | | | |
| Allocation | | X | | | | | | | |
| **INTERVENTIONS:** | | | | | | | | | |
| Radiofrequency | | | ▬▬▬ | | | | | | |
| PFMT | | | ▬▬▬▬▬▬▬▬▬▬▬▬ | | | | | | |
| **ASSESSMENTS:** | | | | | | | | | |
| Sociodemographic Data | | X | | | | | | | |
| Medical History | | X | | | | | | | |
| Obstetric History | | X | | | | | | | |
| PFDI-20 | | X | | | | | X | X | X |
| PFM Strength | | X | | | | | X | X | X |
| ICIQ-SF | | X | | | | | X | X | X |
| Sandvick test | | X | | | | | X | X | X |
| VHI | | X | | | | | X | X | X |
| FSFI | | X | | | | | X | X | X |
| Dispareunia | | X | | | | | X | X | X |
| S-BIS | | X | | | | | X | X | X |
| Adverse Events | | | | | | | X | X | X |

**Fig 1. Description of the stages of the study.** Follow-up (15 days after intervention); F2: Follow-up (6-months after intervention); F3: Follow-up (12-months after intervention); w: week; PFMT: Pelvic Floor Muscle Training; PFDI-20: Pelvic Floor Distress Inventory short form; MSP: ICIQ-SF: International Consultation on Incontinence Questionnaire Short-Form; Vaginal Health Index; FSFI: Female Sexual Function Index; S-BIS: Spanish version of the Body Image Scale.

## Study setting

The recruitment of breast cancer survivors will be carried out at the facilities of two breast cancer associations in the province of Alicante: Association of Women Affected by Breast Cancer of Elche and Region (AMACMEC, Elche) and Vinalopó Breast Cancer Association (AcMAVI, Petrel). Both associations have signed a collaboration agreement with Universidad CEU Cardenal Herrera for the development of the study. The evaluation and intervention will be carried out at the Department of Nursing and Physiotherapy of the Faculty of Health Sciences of Universidad CEU Cardenal Herrera (Elche, Spain).

## Study population

Female breast cancer survivors diagnosed with pelvic dysfunction assessed by ICIQ-SF.

## Sample size

The sample size was calculated according to a previous study that demonstrated statistically significant difference in pelvic floor symptoms and pelvic floor-related quality of life following RF treatment [42]. For sample calculation, the minimum clinically significant effect size of the PFDI-20 was 30 points, with a Cohen's d of 0.612 between extreme groups, assuming a standard deviation of 50, with a 30% loss to follow-up, an alpha risk (α) of 5% and a power (1-β) of 80%. A sample size of 117 women (39 in each group) was obtained using the G*Power program and the one-way ANOVA statistical test with an effect size f: 0.25 (calculated from Cohen's d = 0.612 between extreme groups).

## Eligibility criteria

Women included will be over 18 years, with a clinical history of breast cancer who are members of AMACMEC and AcMAVI, who agree to participate in the study, are willing to attend the treatments on the scheduled dates and at the relevant venues, and who present pelvic dysfunction evaluated by the Pelvic Floor Distress Inventory short form questionnaire (PFDI20) ≥ 100 [43]. This criterion was based on previous results obtained in a cross-sectional observational study conducted by the research group in which pelvic floor-related symptoms were characterised in 250 breast cancer survivors, members of AMACMEC (unpublished data).

Survivors with the following conditions will be excluded from the study: having performed PFMT or having received RF in the last 12 months, use of vaginal oestrogens in the last 6 months, systemic hormone therapy in the last 6 months, laser therapy in the last 6 months, absence of pelvic floor contraction according to the Modified Oxford Scale [44], use of pacemakers, decompensated heart or metabolic diseases, cognitive deficit, peripheral or central neurological disorders, previous pelvic region surgery, skin pathologies or wounds in the treatment area, presence of an active urinary tract and/or vaginal infection or having been diagnosed with another type of cancer or metastasis.

## Recruitment

Participants will be recruited from the associations AMACMEC and AcMAVI between 5 May and 31 July of 2025. For participant recruitment, informative group talks on the project will be held. All participants will be contacted by phone by the researcher who establishes eligibility, while further information related to the project will be provided at a personal interview on the day of the participant's evaluation for inclusion in the study. Participants interested in the study who do not meet the eligibility criteria will be given an initial evaluation, informed of the status of their pelvic function and given therapeutic recommendations, but they will not be able to participate in the study.

## Allocation

The randomisation sequence will be obtained using SAS 9.4 software (SAS Institute, Cary, NC, USA) with a 1:1:1 allocation ratio. The numbers corresponding to the study groups (1. RF and 2. TFMW, 3. RF + PFMT) will be placed in sealed

opaque envelopes that will be opened by the study participants after signing the written informed consent and undergoing an initial evaluation at the first clinical visit.

The researchers who help to complete the questionnaires, perform the physical examination and the measurement of pre- and post-intervention variables, as well as the data analysts, will be blinded to the treatment group to which participants are randomised. It will not be possible to blind the participants regarding the treatment.

### Initial assessment

The recruited patients will be contacted by phone during the informative group talks held at the associations and the first consultation will be scheduled for the initial evaluation. At the initial evaluation, participants will be provided with more information about the study, the interventions, the evaluations, the follow-up, the expected benefits and the risks or discomforts that they may suffer from taking part in the study. Patients will be informed about data protection and their rights when accepting participation. At this point, participants will be able to find answers to all their questions about the study. Participants who agree to participate in the study will receive a consent form to read and sign. Participant personal data will be protected and a number will replace their identities.

At the first visit, participants who accept to participate will complete the PFDI20 questionnaire and an ad hoc questionnaire designed to ascertain sociodemographic characteristics, as well as information related to the diagnosis and treatment of breast cancer, in addition to other specific validated questionnaires to assess pelvic function, vaginal symptoms, sexual function and the perception of body image, which will be further explained later. Pelvic floor muscle strength will then be determined. After the random allocation described above, participants will be informed of the assigned group and the intervention protocol by means of an opaque envelope.

### Interventions

**Group 1. RF-based treatment.** The proposed therapy will consist of applying RF in the modality of capacitive electrical transfer (Capenergy® device model C500 Urogyne, Capenergy Medical, Barcelona, Spain). A device designed mainly for addressing dysfunction of the urogynaecological region, where the increase in tissue temperature is regulated by a temperature sensor, having 3 frequencies (0.8MHz, 1MHz and 1.2 MHz) that will allow addressing different tissue depths, and a power of 310w. This device consists of two electrodes: a capacitive active one that will be placed in the vaginal area with a probe cover and water-soluble gel and another dispersive electrode or return plate that will be positioned in the lumbosacral region.

The protocol is based on a previous treatment that proved to be effective for the treatment of GSM in post-menopausal women [45]. Participants will be placed in the lithotomy position, with the legs flexed and supported. Treatment temperature will be set at 41ºC, with a frequency of 1MHz and power of 75KJ. Once that temperature has been reached, the physiotherapist will perform semi-circular movements on the vaginal wall, on the vaginal anterior side for 2 minutes and on the vaginal posterior wall for 4 minutes. There will be a total number of 5 sessions with a 7-day interval between them, which will total 4 weeks of treatment. Patients will be instructed to contact the researchers if they experience any discomfort or notice any changes in vaginal discharge.

**Group 2. PFMT-based treatment.** PFMT-group patients will be assisted during the sessions by experienced physiotherapists. For PFMT, the participants, in groups of 8 people, will perform a CORE and pelvic floor exercise programme based on the strength, endurance and fatigability assessment of the patients. The goal is to activate the pelvic floor muscles alone and together with the CORE muscles, both statically and dynamically.

The design of the exercise protocol and the sequences is based on an adaptation of the programme described in a previous study carried out in 117 climacteric women with pelvic dysfunction where significant improvement in UI symptoms, vaginal symptoms and sexual function was shown, similar to what is intended to be analysed in this project [39]. Previous studies also considered addressed therapeutic exercise in the prevention and treatment of pelvic floor pathologies using PFMT, hypopressive technique, CORE training and unstable surface training [46–52].

PFMT will be performed twice a week, each session lasting 45 minutes and the total training period being 16 weeks. PFMT will expand objectives as the months progress and sessions will vary each week, to promote adherence and motivation based on the diversity of exercises. The first four weeks will include a first day of CORE and pelvic floor training in a cubicle and an individual session to ensure the understanding of concepts and proper implementation of the technique. Thereafter, the therapeutic exercise protocol will focus on proprioception, mobilisation, and activation of structures responsible for the CORE, isometric work, voluntary activation of the CORE muscles, pelvic floor and synergistic muscles such as the gluteus, both statically and dynamically, as well as performing exercises in resisted expiration and apnoea to facilitate the activation of the lumbopelvic complex, as other programmes targeting patients with abdominopelvic dysfunction have proven to be effective [49, 53-55].

In the second month, mobility, proprioception and PFMT exercises facilitated by posture and breathing will be maintained, and the load will be increased with positions countering gravity, activation in movement and activation against resistance of the accessory muscles [56–58].

The third month will continue with mobility, proprioception and PFMT exercises with increased load through posture and movement, and external loads will be introduced up to 60% of the maximum repetition according to the progression of the loads in terms of strength, endurance and health [59].

In the fourth month, after re-evaluating the maximum repetition for load management, the load will be increased to 75% and we will focus on CORE and pelvic floor training and activation in a dynamic manner, with normalised breathing and introducing impact and fatigue, demonstrable risk factors in pelvic pathology. The objective is to achieve the automation of the abdominopelvic synergy and its competence in situations of both daily life and physical exercise [52, 60].

The detailed protocol with the description of each of the exercises performed on each training day is available in the supporting information (S2 File). In case of two absences, patients will be contacted and their participation in the study will be re-evaluated. The individual pictured in the physical exercise programme description has provided written informed consent (as outlined in PLOS consent form) to publish their image alongside the manuscript.

## Primary outcomes

The primary outcome is improvement in pelvic function following the proposed interventions, assessed in terms of reduction in symptoms and the impact of pelvic dysfunction, including urinary, colorectal-anal, and genital prolapse symptoms.

**Assessment Criteria.** - Tool: Pelvic Floor Distress Inventory Questionnaire, short form (PFDI-20) [43]. The PFDI-20 questionnaire uses a scoring system where each of the 20 items is rated from 0 to 4, with 0 meaning the symptom is not present and 1–4 indicating how much it bothers, from "not at all" to "quite a bit."

- Details: This questionnaire assesses the impact of pelvic dysfunction over the last 3 months and consists of three subscales:

Pelvic Organ Prolapse Impact Questionnaire (POPIQ-7): 6 items (questions 1–6), focusing on distress from pelvic organ prolapse.

Colon-Rectal-Anal Impact Questionnaire (CRAIQ-7): 8 items (questions 7–14), assesses colon-rectal-anal symptoms.

Urinary Impact Questionnaire (UIQ-7): 6 items (questions 15–20), evaluating urinary symptoms.

- Scoring: The maximum score is 300 points, with a maximum of 100 points for each subscale. For each scale, calculate the mean score of the items, then multiply by 25 to get a score from 0 to 100. A higher score indicates a greater negative impact on quality of life, so improvement is reflected in a reduction in the score, indicating fewer or less severe symptoms.

- Assessment Timing: It is measured at baseline and at follow-ups 15 days, 6 months, and 12 months post-intervention.

## Secondary outcomes

Secondary outcomes include a series of assessments related to pelvic health, vaginal symptoms, sexual function, and quality of life, conducted at the same time as the primary outcome (baseline, 15 days, 6 months, and 12 months post-intervention). Each outcome, its definition and description, and the assessment criteria are detailed below:

**Assessment of pelvic floor muscle strength.** Measures the strength, endurance, and fatigability of the pelvic floor muscles, which are essential for supporting the pelvic organs and controlling urinary and faecal continence.

Prior to measurement, participants will be asked to go to the bathroom to urinate, thus allowing to standardise, as far as possible, bladder volume [61]. They will rest for 3 minutes in a sitting position prior to the determination, since this is twice the time required for sympathetic deactivation. To register the strength, the women will be placed in the lithotomy position, with the genital region and legs undressed and covered by a sheet. They will then be instructed to remain relaxed [62].

The first examination will consist of a bidigital palpation to estimate the strength of the pelvic floor muscles during maximum contraction using the Oxford scale and based on previous studies which report the influence of strength, endurance and fatigability on pelvic floor function and its relationship with synergistic muscles. The Oxford scale assesses the contractile capacity of the pelvic floor muscles. Scores range from 0 to 5, as follows: no contraction is rated 0, a very weak contraction is rated 1, a weak contraction is 2, a moderate/tensioned/and sustained contraction is rated 3, a good contraction with sustained tension and resistance is 4, and a strong contraction with sustained tension against a strong resistance is 5 [44]. In addition, this assessment allows to establish the static muscular resistance, the fatigability or dynamic resistance and the maximum muscular strength [44, 63, 64].

The strength and endurance will also be determined by using a pelvimeter consisting of an inflatable vaginal probe connected to the therapeutic device for neuromuscular stimulation and manometry PHENIX LIBERTY (Electronic Concept Lignon Innovation, Montpellier, France) [65]. The air probe, connected to the Phenix biofeedback system, covered by a gel-lubricated latex probe cover will be used. In the procedure, the labia majora are opened with one hand and with the other hand holding the back of the manometric probe, it is slowly rotated into the vagina. Pelvic floor pressure signals will be collected by measuring both its basal tone and the maximum pressure maintained for 10 seconds in three measurements, the average of the three being calculated at the command, "contract as much as you can for as long as possible".

**Pelvic function and quality of life.** Assesses overall pelvic function and its impact on quality of life, beyond the primary outcome, with an emphasis on urinary incontinence and its severity. Pelvic function will be assessed with the PFDI-20 questionnaire described above, and in addition using the following questionnaires:

The International Consultation on Incontinence Questionnaire – Short Form (ICIQ-SF) is a self-administered four-question questionnaire that identifies people with urinary incontinence by assessing the frequency, severity and impact on quality of life. It consists of five questions that evaluate the frequency, severity and impact of UI, in addition to a set of eight self-diagnostic items related to UI situations experienced by patients. The maximum sum of the response values scores 21 points, referring to the high impact of UI on an individual's life [66].

The Sandvik severity test established UI severity based on two questions. The interpretation based on the score is classified as: 1–2 mild UI, 3–6 moderate UI, 8–9 severe UI, 12 very severe UI [67].

**Vaginal symptoms.** Assesses vaginal health, including elasticity, fluid volume, pH, epithelial integrity, and moisture, which are frequently affected in GSM. By physical examination, the vaginal health index (VHI) will be evaluated, which consists of a graduated scale from 1 to 5 for each item (vaginal elasticity, fluid volume, pH, epithelial integrity and humidity) [45].

Vaginal elasticity ranges from 1 (no elasticity) to 5 (excellent elasticity), evaluated by distension of the mucosa on palpation and the placement of the vaginal speculum.

Volume of fluid, evaluated during the examination, varies between 1 (no secretion) and 5 (normal secretion, white, flocculent).

Epithelium integrity ranges from 1 (petechiae detected on examination) to 5 (non-friable tissue and normal mucosa). Humidity ranges from 1 (no humidity is detected on examination and presence of swollen mucosa) to 5 (normal humidity).

The pH will be quantified using a pH indicator strip between 0 and 14 that will be placed directly on the right lateral vaginal wall for one minute, scoring 1 point for a pH of 6.1, 2 for pH 5.6–6.0, 3 for pH 5.1–5.5, 4 for pH 4.7–5.0 and 5 for pH ≤ 4.6.

The sum of all the items represents the vaginal health score, where 25 represents the best vaginal health [68].

**Sexual function and self-esteem.** Assesses sexual function (desire, arousal, lubrication, orgasm, satisfaction, and pain) and perceived body image, which may be affected by breast cancer treatments and pelvic dysfunction. The Female Sexual Function Index (FSFI) questionnaire consists of 19 items and evaluates the sexual function in the last 4 weeks and the performance in six domains: sexual desire, arousal, lubrication, orgasm, satisfaction and pain. The cut-off point ≤ 26.5 is considered sexual dysfunction and an increase in the score is considered an improvement [45].

Dyspareunia will be evaluated using the Visual Analogue Scale (VAS) [69], to measure the pain intensity described by the patient with maximum inter-observer reproducibility. It consists of a horizontal 10-cm line whose ends represent the extreme expressions of a symptom. No intensity or lower intensity is located on the left, while higher intensity is represented on the right. The patient will be asked to mark on the line the point of her pain intensity during sexual intercourse and it is measured with a millimetric ruler. The intensity is expressed in centimetres or millimetres. The assessment will be: 1 Mild pain, if the patient scores the pain as less than 3; 2 Moderate pain, if the score is between 4 and 7; 3 Severe pain, if the score is equal to or greater than 8.

The Body Image Scale (S-BIS) consists of 10 items that evaluate various dimensions of body image in cancer patients, evaluating: affectivity, behaviour and cognitive dimension [70]. Products are rated on a four-point scale (0: none; 1: a little; 2: quite a lot; 3: a lot); the maximum possible score is 3 points. The higher the score obtained, more problematic are the issues related to body image. Its brevity favours a rapid evaluation, both in the clinical and research settings.

**Baseline and follow-up assessment.** After signing the informed consent, patients will be subjected to an anamnesis that includes questions related to a questionnaire designed ad hoc to identify the sociodemographic characteristics, as well as information related to the diagnosis and treatment of breast cancer. They will complete the questionnaires described above about pelvic function and quality of life, vaginal health, sexual function and self-esteem. Subsequently, patients will be referred for a physical examination consisting of a pelvic floor muscle strength evaluation. The questionnaires will be self-administered by the participants and collected at the end of the baseline assessment. These procedures will be performed at the initial assessment (first visit).

Patients will be followed up at three time-points after the end of the interventions: first follow-up (15 days after the intervention), second follow-up (six months after the intervention) and third follow-up (12 months after the intervention). The same evaluation procedures of the baseline follow-up will be carried out in the successive follow-ups.

**Data collection and management.** A researcher will assist the participants in completing the questionnaires and will check that all the questionnaires are completed and signed. A physiotherapist with more than 15 years of experience in pelvic floor treatment will evaluate the strength of the pelvic floor muscles and vaginal health of all participants. Two physiotherapists with more than 15 years of experience in rehabilitation will perform the PFMT interventions. A physiotherapist with more than 10 years of experience will perform the RF intervention.

The data collected for this study will be included in a computerised database of the Universidad CEU Cardenal Herrera, to which access will only be granted to the researcher in charge of the analysis, extraction and handling of the same, through an access keyword, being subject to the secrecy inherent to his or her profession or derived from a non-disclosure agreement.

## Harms

Although RF is a well-tolerated procedure, adverse effects may occur. The RF intervention will be discontinued in the presence of discomfort during its application evaluated with the Visual Analogue Scale, as well as in the case of events such as urinary tract infection, vulvovaginitis, irritation or vaginal injury [71, 72]. In these cases, appropriate medical

treatment will be offered. PFMT is not associated with adverse effects. Women with less than 80% attendance at the RF or PFMT sessions will not be considered to comply with the study protocol and their participation will be terminated, although they will be included in the data analysis (by intention-to-treat).

## Data analysis

The Kolmogorov-Smirnov test will be used to evaluate the normality of the sample.

For the analysis of PFDI-20 scores (continuous variable) that determine improvement in pelvic function (primary outcome), ANOVA will be used if the data follow a normal distribution. If the data are not normally distributed, the Kruskal-Wallis test will be applied for intergroup evaluation in order to compare the three intervention groups (RF, PFMT and RF+PFMT) and determine which intervention is most effective in determining pelvic function. To evaluate changes within each group, the Student's t-test for paired samples will be used if the data are normal, or the Wilcoxon test if they are not, with the aim of evaluating the magnitude of improvement in pelvic function within each group over time. Repeated measures ANOVA to simultaneously analyse the effect between groups, the effect within groups and the group x time interaction to determine whether the effect of the intervention varies over time between groups.

Regarding the analysis of secondary variables, the comparative analysis between groups will be performed using ANOVA if the data are normal, or using the Kruskal-Wallis test if they are not, for continuous variables (Oxford scale for muscle strength, VHI, FSFI and S-BIS). For categorical variables (Sandvick severity test and dyspareunia using VAS), the chi-square test or Fisher's exact test will be used to analyse associations between groups. Intragroup evaluation will be performed using Student's t-test (normal data) or Wilcoxon's test (non-normal data) for continuous secondary variables, and McNemar's test will be applied to evaluate differences before and after the intervention for categorical secondary variables. Similarly, a repeated measures analysis will be performed using ANOVA to simultaneously analyse the effect between groups, the effect within groups and the group x time interaction to capture the short- and long-term effects of the interventions on the secondary outcomes.

The results will be analysed based on the intention-to-treat principle. The significance level will be 5%. IBM SPSS Statistics Windows software, version 29.0 (SPSS Inc., Chicago, IL, USA) will be used. The statistical analysis of the collected data will be performed by a blinded researcher for the intervention and for data collection.

## Ethical considerations

The present study has been approved by the Ethics Committee for Biomedical Research of the Universidad CEU Cardenal Herrera (CEEI24/540). This study is also registered at ClinicalTrials.gov as a clinical trial (19/11/2024, NCT06694519). The original version presented to the Ethics Committee in Spanish and its English translation are provided as supplementary information (S3 File and S4 File).

The study will be carried out according to the Declaration of Helsinki and current Spanish legislation (Royal Decree 223/2004 and Biomedical Research Law of 2007). Participants who agree to participate in the study will receive a consent form to read and sign. The blinding of the registered results will be guaranteed since no personal data that allow subject identification or data referring to email accounts will be collected. The data will be processed in accordance with the provisions of Regulation (EU) 2016/679 of the European Parliament and of the Council of 27 April 2016 on the protection of individuals with regard to the processing of personal data and on the free movement of such data and repealing Directive 95/46/EC. The individual in this manuscript has given written informed consent (as outlined in PLOS consent form) to publish these case details.

## Trial status

On 5 May 2025, information talks and recruitment of participants for the study will begin. The participant recruitment will be completed on 31 July 2025. The start of the intervention is scheduled for July 2025 and data collection will be completed in January 2027. Final results are expected in April 2027.

## Discussion

GSM-associated pelvic dysfunction is a barely addressed problem among breast cancer survivors and the absence of efficient evidence-based conservative treatments negatively affects the management of this condition.

The scientific-technical impact of this project lies in the fact that it will provide scientific evidence of the effect of a conservative and non-invasive treatment such as RF as a single treatment or adjuvant to physical exercise, a first-choice treatment. Based on the study outcomes, it will be possible to update the treatment protocols for the prevention and management of pelvic floor dysfunction in breast cancer survivors and increase patients' quality of life. All this will contribute to expanding the scientific evidence of the performance of physiotherapy in the field of oncology, favouring the development of new lines of research in both RF and therapeutic physical exercise in patients with other types of oncological conditions, as well as in patients with genitourinary syndrome derived from both menopause and the use of treatments and/or interventions that lead to it.

This study will promote the development of health promotion campaigns, raising awareness in health professionals themselves of the need for breast cancer survivors to be able to prevent the onset of pelvic floor dysfunction, mainly derived from the surgical and pharmacological treatments to which they are subjected.

### Limitations of the study design

The main limitation of the study lies in achieving adherence to the PFMT protocol. A sensitivity analysis will be carried out to manage attrition bias and the processing of lost data. The results will be analysed by intention-to-treat.

Another important limitation of the present study is the non-inclusion of a control group, justified by ethical considerations given the need to treat all participants.

On the other hand, there may be some variability in training intensity, so the researchers will be instructed to constantly monitor the effort perceived by the study subjects when performing the physical exercise programme.

### Dissemination plans

The expected results of the study are part of a doctoral thesis and will be disseminated at specialised national and international scientific congresses, as well as in specific journals with the highest possible impact factor according to Journal Citation Reports.

### Study amendments

If the protocol needs to be modified, this modification will be communicated to the Ethics Committee for Biomedical Research, clinical trial registry, the journal and the funding agency of the study.

### Supporting information

**S1 File. Checklist.** SPIRIT 2013 checklist study protocol.
(DOCX)

**S2 File. PFMT Programme.**
(DOCX)

**S3 File. Report English version.**
(DOCX)

**S4 File. Report Spanish version.**
(DOCX)

## Acknowledgments

The authors would like to thank the Asociación de Mujeres Afectadas por el Cáncer de Mama de Elche y Comarca (Breast Cancer Association of Elche and Region) (AMACMEC, Elche, Spain) and Asociación cáncer de Mama Vinalopó (Vinalopó Breast Cancer Association) (AcMAVI, Petrel, Spain) for making the facilities available to us for the dissemination of the protocol among the Associations' breast cancer survivors. We thank the Department of Nursing and Physiotherapy of the Faculty of Health Sciences of the Universidad CEU Cardenal Herrera for offering its facilities for data collection and interventions. We thank the Research Ethics Committee of the Universidad CEU Cardenal Herrera for its guidance in the design and validation of the study protocol. Special thanks to Capenergy Medical (Barcelona, Spain) for collaborating in the study with the provision of the RF devices.

## Author contributions

**Conceptualization:** Cristina Salar-Andreu, Sergio Montero-Navarro, María Torres-Lacomba, Josep C. Benítez-Martínez, Jesús Sánchez-Más, Cristina Orts-Ruiz.

**Funding acquisition:** Cristina Salar-Andreu, Cristina Orts-Ruiz.

**Methodology:** Cristina Salar-Andreu, Sergio Montero-Navarro, Ana Lozano-Rubio, Sonia Del Rio-Medina, Jose M. Botella-Rico, Josep C. Benítez-Martínez.

**Project administration:** Jesús Sánchez-Más.

**Supervision:** Cristina Salar-Andreu, María Torres-Lacomba, Josep C. Benítez-Martínez, Jesús Sánchez-Más, Cristina Orts-Ruiz.

**Validation:** Cristina Salar-Andreu, Sergio Montero-Navarro, Ana Lozano-Rubio, Sonia Del Rio-Medina, Jose M. Botella-Rico, María Torres-Lacomba, Josep C. Benítez-Martínez, Jesús Sánchez-Más, Cristina Orts-Ruiz.

**Visualization:** Cristina Salar-Andreu, Sergio Montero-Navarro, Cristina Orts-Ruiz.

**Writing – original draft:** Cristina Salar-Andreu, Sergio Montero-Navarro, Sonia Del Rio-Medina, Jose M. Botella-Rico, María Torres-Lacomba, Josep C. Benítez-Martínez, Jesús Sánchez-Más, Cristina Orts-Ruiz.

**Writing – review & editing:** Cristina Salar-Andreu, Sergio Montero-Navarro, Sonia Del Rio-Medina, Jose M. Botella-Rico, María Torres-Lacomba, Josep C. Benítez-Martínez, Jesús Sánchez-Más, Cristina Orts-Ruiz.

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
