## [Decision Letter · Decision Letter 0]

19 Jun 2025

PONE-D-25-12125Effectiveness of radiofrequency and exercise-based rehabilitation on symptoms associated with pelvic floor dysfunction in breast cancer patients: A study protocol.PLOS ONE

Dear Dr. Sanchez Mas,

Thank you for submitting your manuscript to PLOS ONE. After careful consideration, we feel that it has merit but does not fully meet PLOS ONE’s publication criteria as it currently stands. Therefore, we invite you to submit a revised version of the manuscript that addresses the points raised during the review process.

We look forward to receiving your revised manuscript.

Kind regards,

Satyajeet Rath

Academic Editor

PLOS ONE

Journal Requirements:

“The authors declare that there are competing financial interests in relation to the described work associated with the funding received by the company Capenergy Medical (Barcelona, Spain).

Capenergy Medical (Barcelona, Spain) will assign the devices for applying RF with capacitive electrical transfer (Capernergy® device model C500 Urogyne) and will finance a three-year scholarship to hire a predoctoral researcher who will participate in the project.”

We note that you received funding from a commercial source: [Name of Company]

“Capenergy Medical (Barcelona, Spain) will assign the devices for applying RF with capacitive electrical transfer (Capernergy® device model C500 Urogyne) and will finance a three-year scholarship to hire a predoctoral researcher who will participate in the project. “

5. Please include a caption for figure 1.

6. We note that the original protocol file you uploaded contains a confidentiality notice indicating that the protocol may not be shared publicly or be published. Please note, however, that the PLOS Editorial Policy requires that the original protocol be published alongside your manuscript in the event of acceptance. Please note that should your paper be accepted, all content including the protocol will be published under the Creative Commons Attribution (CC BY) 4.0 license, which means that it will be freely available online, and any third party is permitted to access, download, copy, distribute, and use these materials in any way, even commercially, with proper attribution.

Therefore, we ask that you please seek permission from the study sponsor or body imposing the restriction on sharing this document to publish this protocol under CC BY 4.0 if your work is accepted. We kindly ask that you upload a formal statement signed by an institutional representative clarifying whether you will be able to comply with this policy. Additionally, please upload a clean copy of the protocol with the confidentiality notice (and any copyrighted institutional logos or signatures) removed.

7. We note that the original protocol that you have uploaded as a Supporting Information file contains an institutional logo. As this logo is likely copyrighted, we ask that you please remove it from this file and upload an updated version upon resubmission.

8 We note that Supporting Information 2 PFMT Programme includes an image of a [patient / participant / in the study].

Please respond by return e-mail with an amended manuscript. We can upload this to your submission on your behalf.

If you are unable to obtain consent from the subject of the photograph, please either instruct us to remove the figure or supply a replacement figure by return e-mail for which you hold the relevant copyright permissions and subject consents. In some cases, you may need to specify in the text that the image used in the figure is not the original image used in the study, but a similar image used for illustrative purposes only. We can make any changes on your behalf.

Additional Editor Comments (if provided):

The reviewers have reviewed the paper and recommended revisions. Kindly undertake the required revision and resubmit it for review.

Reviewers' comments:

Reviewer's Responses to Questions

**Comments to the Author**

1. Does the manuscript provide a valid rationale for the proposed study, with clearly identified and justified research questions?

Reviewer #1: Yes

Reviewer #2: Yes

2. Is the protocol technically sound and planned in a manner that will lead to a meaningful outcome and allow testing the stated hypotheses?

Reviewer #1: Yes

Reviewer #2: Yes

3. Is the methodology feasible and described in sufficient detail to allow the work to be replicable?

Reviewer #1: No

Reviewer #2: Yes

4. Have the authors described where all data underlying the findings will be made available when the study is complete?

Reviewer #1: Yes

Reviewer #2: Yes

5. Is the manuscript presented in an intelligible fashion and written in standard English?

Reviewer #1: Yes

Reviewer #2: Yes

6. Review Comments to the Author

You may also provide optional suggestions and comments to authors that they might find helpful in planning their study.

Reviewer #1: The authors present a protocol for a triple arm randomised clinical trial to improve the quality of life of breast cancer patients. The research objectives are good but the protocol needs some methodological/statistical improvement.

1. Abstract: Methods: All clinical trials are prospective and hence the word prospective is not needed.

2. Sample size needs to be calculated and presented based on the assumed minimum clinically worthwhile effect size in terms of primary end point. Also, the test statistics behind the sample size and the software used for calculation needs to be mentioned.

3. Primary outcome and secondary outcomes are not clearly defined. Primary outcome and secondary outcomes need to be clearly defined with their corresponding endpoints.

4. Data analysis section needs to be stratified in terms of primary endpoint analysis and secondary endpoint analysis.

Reviewer #2: The Manuscript is a trial protocol only, possibly a part of doctoral thesis. Rationale of the study, inclusion criteria, intervention methods, statistical methodology of proposed data analysis is given in details, but results are not given it being just a trial protocol.

The limitations of the study are clearly identified as well.

The PI are requested to give more details of the scoring system used for evaluation of primary outcome. (PFDI-20 questionnaire and its scoring methods and calculations).

7. PLOS authors have the option to publish the peer review history of their article (what does this mean? ). If published, this will include your full peer review and any attached files.

**Do you want your identity to be public for this peer review?** For information about this choice, including consent withdrawal, please see our Privacy Policy .

Reviewer #1: **Yes: ** Dr Shah-Jalal Sarker

Reviewer #2: No

---

## [Author Response · Author response to Decision Letter 1]

4 Jul 2025

Below, I include the cover letter in which I respond in detail to each of the editor's instructions in the email we received. This information is also attached as a document named "Cover Letter reviewed".

I also include a point-by-point response to Reviewers comments. This information is also attached as a document named "Response to reviewer".

COVER LETTER REVIEWED

All changes are indicated in red in the manuscript reviewed with track changes.

Dear Editor, please find attached the manuscript entitled “Effectiveness of radiofrequency and exercise-based rehabilitation on symptoms associated with pelvic floor dysfunction in breast cancer patients: a study protocol”.

We are grateful for all the contributions that have improved the writing and presentation of the findings, as well as contributed to increase the quality of the study presented.

Here follows a point-by-point response to your comments. All changes are indicated in red in the manuscript.

Editor Comments

Journal Requirements:

The PLOS ONE's style requirements have been reviewed in the manuscript and modified.

“The authors declare that there are competing financial interests in relation to the described work associated with the funding received by the company Capenergy Medical (Barcelona, Spain). Capenergy Medical (Barcelona, Spain) will assign the devices for applying RF with capacitive electrical transfer (Capernergy® device model C500 Urogyne) and will finance a three-year scholarship to hire a predoctoral researcher who will participate in the project.”

We note that you received funding from a commercial source: [Name of Company]

Thank you for your suggestions. A conflict of interests statement from the International Committee of Medical Journal Editors has been included as an attachment to the submission (coi-disclosure). In addition, the description in the manuscript and cover letter has been modified as follows:

“The authors declare that there are competing financial interests in relation to the described work associated with the funding received by the company Capenergy Medical (Barcelona, Spain). Capenergy Medical (Barcelona, Spain) will assign the device for applying RF with capacitive electrical transfer (Capernergy® device model C500 Urogyne). This does not alter our adherence to PLOS ONE policies on sharing data and materials. There are no other conflicts of interest.”

“Capenergy Medical (Barcelona, Spain) will assign the devices for applying RF with capacitive electrical transfer (Capernergy® device model C500 Urogyne) and will finance a three-year scholarship to hire a predoctoral researcher who will participate in the project. “

The amended statements have been included in the cover letter. The information regarding funding has been removed from the manuscript. The funding statement remains as stated in the online submission, ‘The author(s) received no specific funding for this work.’, as no funding has been received for the publication of this study protocol.

The manuscript has been revised, and the ethical statement now appears only in the Methods section. Thank you.

5. Please include a caption for figure 1.

Figure 1 has been submitted in TIF format as an attachment. The title and legend have been included in the manuscript where they should be, in accordance with PLOS ONE's style requirements.

6. We note that the original protocol file you uploaded contains a confidentiality notice indicating that the protocol may not be shared publicly or be published. Please note, however, that the PLOS Editorial Policy requires that the original protocol be published alongside your manuscript in the event of acceptance. Please note that should your paper be accepted, all content including the protocol will be published under the Creative Commons Attribution (CC BY) 4.0 license, which means that it will be freely available online, and any third party is permitted to access, download, copy, distribute, and use these materials in any way, even commercially, with proper attribution.

Therefore, we ask that you please seek permission from the study sponsor or body imposing the restriction on sharing this document to publish this protocol under CC BY 4.0 if your work is accepted. We kindly ask that you upload a formal statement signed by an institutional representative clarifying whether you will be able to comply with this policy. Additionally, please upload a clean copy of the protocol with the confidentiality notice (and any copyrighted institutional logos or signatures) removed.

We did not find any clause in the original file approved by the Ethics Committee that indicates that the protocol cannot be shared or published. The only confidentiality described refers to personal data that could identify participants, which in that case must be processed confidentially as indicated in the protocol.

If you could give me a more specific indication of this confidentiality notice you mention, I could modify it without any problem, but we cannot find it in the original protocol. I assure you that there is no objection or clause preventing the publication of the protocol. Thank you very much, and we look forward to receiving your instructions if we are mistaken.

7. We note that the original protocol that you have uploaded as a Supporting Information file contains an institutional logo. As this logo is likely copyrighted, we ask that you please remove it from this file and upload an updated version upon resubmission.

The institutional logo has been removed. Thank you.

8 We note that Supporting Information 2 PFMT Programme includes an image of a [patient / participant / in the study].

As per the PLOS ONE policy (http://journals.plos.org/plosone/s/submission-guidelines#loc-human-subjects-research ) on papers that include identifying, or potentially identifying, information, the individual(s) or parent(s)/guardian(s) must be informed of the terms of the PLOS open-access (CC-BY) license and provide specific permission for publication of these details under the terms of this license. Please download the Consent Form for Publication in a PLOS Journal (http://journals.plos.org/plosone/s/file?id=8ce6/plos-consent-form-english.pdf ). The signed consent form should not be submitted with the manuscript, but should be securely filed in the individual's case notes. Please amend the methods section and ethics statement of the manuscript to explicitly state that the patient/participant has provided consent for publication: “The individual in this manuscript has given written informed consent (as outlined in PLOS consent form) to publish these case details”.

Please respond by return e-mail with an amended manuscript. We can upload this to your submission on your behalf.

Following your instructions, the consent forms have been downloaded and authorisation has been obtained from the two individuals who appear in the PFMT programme (supplementary material S2). The signed forms have been filed. This information has been included in the ethical statement in the Methods section. If you need us to send you the signed forms, please let us know.

RESPONSE TO REVIEWERS

Reviewer Comments:

Reviewer #1: The authors present a protocol for a triple arm randomised clinical trial to improve the quality of life of breast cancer patients. The research objectives are good but the protocol needs some methodological/statistical improvement.

1. Abstract: Methods: All clinical trials are prospective and hence the word prospective is not needed.

The word ‘prospective’ has been removed from the abstract and from the Trial Design section of the Materials and Methods. Thank you.

2. Sample size needs to be calculated and presented based on the assumed minimum clinically worthwhile effect size in terms of primary end point. Also, the test statistics behind the sample size and the software used for calculation needs to be mentioned.

The information requested by the reviewer has been included in the section “Study population” of the Materials and Methods (page 7) as follows:

For sample calculation, the minimum clinically significant effect size of the PFDI-20 was 30 points, with a Cohen's d of 0.612 between extreme groups, assuming a standard deviation of 50, with a 30% loss to follow-up, an alpha risk (α) of 5% and a power (1-β) of 80%. A sample size of 117 women (39 in each group) was obtained using the G*Power program.

3. Primary outcome and secondary outcomes are not clearly defined. Primary outcome and secondary outcomes need to be clearly defined with their corresponding endpoints.

We appreciate this contribution from the reviewer. The main results have been developed in more detail in the section ‘Materials and methods’ (pages 11 and 12), as follows:

Primary outcomes

The primary outcome is improvement in pelvic function following the proposed interventions, assessed in terms of reduction in symptoms and the impact of pelvic dysfunction, including urinary, colorectal-anal, and genital prolapse symptoms.

Assessment Criteria:

- Tool: Pelvic Floor Distress Inventory Questionnaire, short form (PFDI-20) 43. The PFDI-20 questionnaire uses a scoring system where each of the 20 items is rated from 0 to 4, with 0 meaning the symptom is not present and 1 to 4 indicating how much it bothers, from "not at all" to "quite a bit."

- Details: This questionnaire assesses the impact of pelvic dysfunction over the last 3 months and consists of three subscales:

Pelvic Organ Prolapse Impact Questionnaire (POPIQ-7): 6 items (questions 1-6), focusing on distress from pelvic organ prolapse.

Colon-Rectal-Anal Impact Questionnaire (CRAIQ-7): 8 items (questions 7-14), assesses colon-rectal-anal symptoms.

Urinary Impact Questionnaire (UIQ-7): 6 items (questions 15-20), evaluating urinary symptoms.

- Scoring: The maximum score is 300 points, with a maximum of 100 points for each subscale. A higher score indicates a greater negative impact on quality of life, so improvement is reflected in a reduction in the score, indicating fewer or less severe symptoms.

- Assessment Timing: It is measured at baseline and at follow-ups 15 days, 6 months, and 12 months post-intervention.

The secondary outcomes have been also expanded with a definition at the beginning of each secondary outcome, and it has been verified that the assessment criteria have been described: tools, procedure, and scoring.

All changes have been indicated in red in the manuscript.

4. Data análisis section needs to be stratified in terms of primary endpoint análisis and secondary endpoint análisis.

Following your recommendation, we have included the following text, stratifying it in terms of primary endpoint analysis and secondary endpoint analysis (page 17):

For the analysis of PFDI-20 scores (continuous variable) that determine improvement in pelvic function (primary outcome), ANOVA will be used if the data follow a normal distribution. If the data are not normally distributed, the Kruskal-Wallis test will be applied for intergroup evaluation in order to compare the three intervention groups (RF, PFMT and RF+PFMT) and determine which intervention is most effective in determining pelvic function. To evaluate changes within each group, the Student's t-test for paired samples will be used if the data are normal, or the Wilcoxon test if they are not, with the aim of evaluating the magnitude of improvement in pelvic function within each group over time. A repeated measures analysis will be performed using ANOVA to simultaneously analyse the effect between groups, the effect within groups and the group x time interaction to determine whether the effect of the intervention varies over time between groups.

Regarding the analysis of secondary variables, the comparative analysis between groups will be performed using ANOVA if the data are normal, or using the Kruskal-Wallis test if they are not, for continuous variables (Oxford scale for muscle strength, VHI, FSFI and S-BIS). For categorical variables (Sandvick severity test and dyspareunia using VAS), the chi-square test or Fisher's exact test will be used to analyse associations between groups. Intragroup evaluation will be performed using Student's t-test (normal data) or Wilcoxon's test (non-normal data) for continuous secondary variables, and McNemar's test will be applied to evaluate differences before and after the intervention for categorical secondary variables. Similarly, a repeated measures analysis will be performed using ANOVA to simultaneously analyse the effect between groups, the effect within groups and the group x time interaction to capture the short- and long-term effects of the interventions on the secondary outcomes.

Reviewer #2: The Manuscript is a trial protocol only, possibly a part of doctoral thesis. Rationale of the study, inclusion criteria, intervention methods, statistical methodology of proposed data analysis is given in details, but results are not given it being just a trial protocol.

The limitations of the study are clearly identified as well.

The PI are requested to give more details of the scoring system used for evaluation of primary outcome. (PFDI-20 questionnaire and its scoring methods and calculations).

We appreciate the reviewer's input. Further details on the scoring and calculation method for the PFDI-20 questionnaire have been provided (pages 11 and 12), as follows:

Primary outcomes

The primary outcome is improvement in pelvic function following the proposed interventions, assessed in terms of reduction in symptoms and the impact of pelvic dysfunction, including urinary, colorectal-anal, and genital prolapse symptoms.

Assessment Criteria:

- Tool: Pelvic Floor Distress Inventory Questionnaire, short form (PFDI-20) 43. The PFDI-20

---

## [Decision Letter · Decision Letter 1]

23 Jul 2025

PONE-D-25-12125R1Effectiveness of radiofrequency and exercise-based rehabilitation on symptoms associated with pelvic floor dysfunction in breast cancer patients: a study protocol.PLOS ONE

Dear Dr. Sanchez Mas,

Thank you for submitting your manuscript to PLOS ONE. After careful consideration, we feel that it has merit but does not fully meet PLOS ONE’s publication criteria as it currently stands. Therefore, we invite you to submit a revised version of the manuscript that addresses the points raised during the review process.

**ACADEMIC EDITOR: Minor Revision**==============================

We look forward to receiving your revised manuscript.

Kind regards,

Satyajeet Rath

Academic Editor

PLOS ONE

Journal Requirements:

Additional Editor Comments:

This article requires minor revision. Kindly do the minor modifications required.

Reviewers' comments:

Reviewer's Responses to Questions

**Comments to the Author**

1. Does the manuscript provide a valid rationale for the proposed study, with clearly identified and justified research questions?

Reviewer #1: Yes

2. Is the protocol technically sound and planned in a manner that will lead to a meaningful outcome and allow testing the stated hypotheses?

Reviewer #1: Yes

3. Is the methodology feasible and described in sufficient detail to allow the work to be replicable?

Reviewer #1: Yes

4. Have the authors described where all data underlying the findings will be made available when the study is complete?

Reviewer #1: Yes

5. Is the manuscript presented in an intelligible fashion and written in standard English?

Reviewer #1: Yes

6. Review Comments to the Author

You may also provide optional suggestions and comments to authors that they might find helpful in planning their study.

Reviewer #1: The software used to calculate the sample size "G*Power" is mentioned by the test statistics behind the sample size calculation is still missing. Ideally, the test statistics used to calculate the sample size needs to be used (and mentioned) to analyse the primary end point.

The sentence, "A repeated measures analysis will be performed using ANOVA" can be shortened by writing, "repeated measures ANOVA".

7. PLOS authors have the option to publish the peer review history of their article (what does this mean? ). If published, this will include your full peer review and any attached files.

**Do you want your identity to be public for this peer review?** For information about this choice, including consent withdrawal, please see our Privacy Policy .

Reviewer #1: **Yes: ** Dr Shah-Jalal Sarker

---

## [Author Response · Author response to Decision Letter 2]

24 Jul 2025

We wish to thank you greatly for the interest you have shown towards our study and for taking the time to review it.

All changes are indicated in red in the “Revised Manuscript with Track Changes”.

Here follows a point-by-point response to your comments:

Reviewer Comments:

Reviewer #1: The software used to calculate the sample size "G*Power" is mentioned by the test statistics behind the sample size calculation is still missing. Ideally, the test statistics used to calculate the sample size needs to be used (and mentioned) to analyse the primary end point.

The information requested by the reviewer has been included in the section “Sample size” of the Materials and Methods (page 9) as follows:

A sample size of 117 women (39 in each group) was obtained using the G*Power program and the one-way ANOVA statistical test with an effect size f: 0.25 (calculated from Cohen's d = 0.612 between extreme groups).

- The sentence, "A repeated measures analysis will be performed using ANOVA" can be shortened by writing, "repeated measures ANOVA".

The change suggested by the reviewer has been made.

---

## [Editor Report · Decision Letter 2]

29 Jul 2025

Effectiveness of radiofrequency and exercise-based rehabilitation on symptoms associated with pelvic floor dysfunction in breast cancer patients: a study protocol.

PONE-D-25-12125R2

Dear Dr. Sanchez Mas,

We’re pleased to inform you that your manuscript has been judged scientifically suitable for publication and will be formally accepted for publication once it meets all outstanding technical requirements.

Kind regards,

Satyajeet Rath

Academic Editor

PLOS ONE

Additional Editor Comments (optional):

As the changes have been made, we accept the paper. Congratulations on your acceptance.